# A Mental Health Pandemic? Assessing the Impact of COVID-19 on Young People’s Mental Health

**DOI:** 10.3390/ijerph20166550

**Published:** 2023-08-09

**Authors:** Katrina Lloyd, Dirk Schubotz, Rosellen Roche, Joel Manzi, Martina McKnight

**Affiliations:** 1School of Social Sciences, Education and Social Work, Queen’s University Belfast, Belfast BT7 1NN, UK; d.schubotz@qub.ac.uk (D.S.); martina.mcknight@qub.ac.uk (M.M.); 2Department of Primary Care, Ohio University HCOM Cleveland Campus, South Pointe Hospital, Warrensville Heights, OH 44122, USA; rocher1@ohio.edu; 3Department of Family and Community Medicine, University of Alabama, Birmingham-Cahaba Medical Care, Birmingham, AL 35211, USA; jamanzi@uabmc.edu

**Keywords:** COVID-19, mental health, young people, Northern Ireland

## Abstract

Background: Research indicates that young people have been a particularly vulnerable group when it comes to negative mental health outcomes following COVID-19, with some authors warning of a ‘mental health pandemic’. Materials and Method: Using a survey approach, this study explored the effects of lockdowns on the mental health of 1995 16-year-olds in Northern Ireland. Respondents completed the 12-item version of the General Health Questionnaire (GHQ-12) along with closed- and open-ended questions about COVID-19. Results: Results from regression analysis showed that being female, identifying as non-heterosexual and perceiving that mental health had worsened during lockdown were the best predictors of poor mental health. In the open responses, young people reported significant concerns about their mental health and their educational outcomes. They also felt that their needs were not given the same priority as those of adults during lockdown. Conclusions: The results suggest that the COVID-19 lockdowns adversely affected the mental health of many young people in Northern Ireland with the effects most acute for females and those identifying as non-heterosexual. Future research should explore the longer-term impact of the pandemic on the mental health of these vulnerable young people and identify what support mechanisms need to be put in place to mitigate the negative effects of any future crises.

## 1. Introduction

The year 2020 was notable for the outbreak and global spread of Coronavirus (COVID-19), an infectious disease caused by the SARS-CoV-2 virus. COVID-19, which can cause severe acute respiratory symptoms in anyone infected, has had a profound impact on the health of the world’s population resulting in around 6.89 million deaths worldwide [1]. In an attempt to curb the spread of COVID-19, most governments across the world implemented ‘lockdowns’ with severe restrictions on social contact and limited access to workplaces and services. Research has suggested that the restrictions imposed during COVID-19 have affected the livelihoods of many people across the globe and the predictions are for a long-term negative impact on employment levels and incomes [2,3]. In addition to the economic and physical health effects of COVID-19, a number of studies have shown that the pandemic has negatively affected the mental health of many people in countries across the world, resulting in higher levels of depression and anxiety, and poorer wellbeing [4,5,6,7,8,9].

However, there is growing evidence that the detrimental effects of the COVID-19 lockdowns on mental health are most notable among particular groups in the population [8,10,11,12] of young adults, women, those identifying as non-heterosexual, people from disadvantaged backgrounds and people with pre-existing mental health problems [10,11,13,14]. For example, across a range of mental health issues, including depression and anxiety, younger people appear to have been affected much more than older people [5,8,13,14,15]. This is perhaps unsurprising since pre-pandemic research had indicated that mental health and wellbeing begin to deteriorate significantly when children enter adolescence [16,17,18,19,20,21], especially among females [19,20,21,22,23,24]. Indeed, as part of a review into the development and risk factors for the onset of social-emotional disorders in young people, a research team concluded that ‘cumulative onset data show a dramatic increase in rates of depression around the period of 14–16 years, especially among girls’ [25] (p. 2).

In an attempt to understand the persistent gender differences in mental ill-health during COVID-19, data collected as part of Understanding Society (US), a longitudinal study of a random sample of households in the UK, were analysed [26]. The findings suggest that among adults aged 16 years and over, social factors (lack of contact with close friends and loneliness) were more salient for women than for men and explained more of the gender difference in mental ill-health than economic factors. Similarly, a study of young people aged between 13 and 16 years found that females experienced a greater decline in mental health during COVID-19 than males [27]. This was attributed to the lack of social contact with friends that the authors suggest has been shown to help support females through times of stressful life events.

A lack of support during COVID-19 lockdowns has also been proposed as one explanation for the higher rate of mental ill-health found among populations identifying as non-heterosexual compared to their heterosexual counterparts [12,28,29]. The vulnerability of this marginalised group in terms of pre-pandemic mental health has been well-documented as has their increased risk of social exclusion and violence as a result of COVID-19 [29,30,31]. For example, research carried out among adults, pre-pandemic, showed a higher rate of prevalence of common mental disorders among those identifying as non-heterosexual [31]. Similarly, a mental health prevalence survey of children and young people in England found that among the 14 to 19 age group, those who identified as lesbian, gay, bisexual or with another sexual identity (LGBTQ+) were more likely to have a mental disorder (34.9%) than those who identified as heterosexual (13.2%) [32]. In one of the largest international studies undertaken on the impact of COVID-19, involving approximately 27,000 participants aged 8 to 17 years in 137 countries, researchers found that the most negative impact of the pandemic was felt by children and young people who were already vulnerable to breaches of their rights before the outbreak of COVID-19 [33]. For example, migrant children and those who identified as LGBTQ+ felt less safe at home than before the pandemic, causing concern for their safety and wellbeing.

Particular concerns, pre-pandemic, had been raised about the mental health and wellbeing of young people in the United Kingdom (UK). This followed the publication of UNICEF Report Card 7 which found that young people in the UK had the poorest wellbeing of the 21 nations of the industrialised world included in the analysis [34]. In 2011, the UK government began to monitor the wellbeing of the nation [35] based mainly, for those over 16 years, on longitudinal data from the US study measured by the GHQ-12, which is a screening instrument for non-psychotic psychiatric disorder widely used in population settings [36,37]. The evidence suggests that, overall, the mental health of the population has been declining since 2014/15 with the largest fall for those aged 16–24 years [8]. Similar findings have been reported for Northern Ireland with time-series data from the Health Survey showing that mental health has been deteriorating (albeit with some fluctuations year on year), especially for the 16–24 age group—14% of this age group were identified as having poor mental health in 2010/11 and this had risen to 23% in 2019/20. The figure for males rose from 8% to 18% and for females from 21% to 28% between 2010/11 and 2019/20 [38]. The figures for the period covering COVID-19 (2020/21) indicate a further rise in cases among this age group (27%). However, the sample size for 2020/21 was too small for analysis by gender. Nonetheless, the findings from Northern Ireland are in line with evidence from the US study, which shows that there has been a rise in poor mental health for all 16–24-year-olds across the UK from 24.5% (2018/19) to 36.7% (April 2020) [8].

Taken together, the evidence points to a worsening of young people’s mental health as a result of COVID-19, particularly among groups who have been identified as vulnerable to mental health issues—namely, young people, females, those identifying as non-heterosexual and disadvantaged groups. However, previous research on the effects of COVID-19 on mental health has been limited by relatively small sample sizes of young people within the teenage years. The current study addresses that gap by analysing data from a sample of 1995 16-year-olds living in Northern Ireland. The aim of the study was to investigate the effects of COVID-19 on the mental health of these young people. Statistical analyses were used to identify the best predictors of poor mental health while responses to open-ended questions provided a more nuanced account of the lived experiences of this cohort of young people, who faced a sudden disruption to their lives during a key stage in their transition to adulthood.

## 2. Methods

YLT is an annual cross-sectional social attitude survey undertaken among 16-year-olds in Northern Ireland since 2003. The survey has been increasingly used by policy makers to monitor young people’s attitudes to, and experiences of, a variety of issues. During the COVID-19 pandemic, ARK was funded by the Office of Northern Ireland Commissioner for Children and Young People (NICCY) to explore how children and young people were affected by the pandemic. YLT is the only annual survey of young people in Northern Ireland with a random sample frame. Ethical approval for the survey was granted by the Research Ethics Committee of the School of Social Sciences, Education and Social Work at Queen’s University Belfast.

Fieldwork for the 2020/21 YLT survey took place in May 2021. Data were collected using a bespoke online survey tool. Optional postal or phone completion was also offered; however, only online responses were received in that year. A sample of 5000 randomly selected 16-year-olds, who lived in Northern Ireland at the time of the fieldwork, was drawn from the Child Benefit Register which is administered by HM Revenue and Customs in the UK. Child Benefit is a social benefit paid to parents or carers who raise children in the UK and is paid for each child. Child Benefit was a universal benefit until March 2013. Then, legislation came into place which introduced means testing with regard to Child Benefit payments. Whilst higher earners with salaries over GBP 50,000 per annum no longer receive Child Benefit, in reality, the Benefit is still paid out, but is then reclaimed from higher earners via income tax (see www.gov.uk/child-benefit (accessed on 30 July 2023)). Whilst we acknowledge that some parents/carers may choose not to apply to receive Child Benefit payments (regardless of their income), due to the way it is administered, the Child Benefit Register remains, de facto, an almost universal sampling frame and the most suitable one for 16-year-olds in Northern Ireland.

Study information sheets with background information about YLT were posted out to the home addresses of eligible 16-year-olds along with instructions on how to participate using an allocated unique ID number, or how to opt out of the study. Informed consent was therefore in place for all survey participants. Participants could choose where and when they completed the survey and what device (smart phone, tablet or computer) they used. In the information letter, 16-year-olds were assured of confidentiality, and they received a link to our privacy notice, which sets out how participant data are used, stored and ultimately destroyed. One reminder letter was sent approximately 10 days later to those who had not yet responded or opted out of the survey by that stage. Every respondent who completed the survey received a GBP 10 gift voucher.

Alongside some background and demographic questions, each year’s YLT survey contains sets of survey questions focusing on various social issues. Themed question modules are funded by various organisations which means different questions are asked each year; however, some questions are included every year to provide a time-series of attitudes. The 2020/21 survey included questions on mental health (assessed using the GHQ-12) and young people’s experiences during COVID-19 lockdowns.

After data cleaning and removal of respondents whose information letters were returned marked as ‘addressee unknown/gone away’, an overall sample of 4913 eligible 16-year-olds remained. A total of 2147 16-year-olds logged onto the survey and 2069 completed all or most questions, giving the survey a response rate of 42.2%. This paper uses data from the 1995 respondents who completed all the items on the mental health measure.

### 2.1. The 12-Item Version of the General Health Questionnaire (GHQ-12)

Mental health data were collected via the GHQ-12, which has been widely used in adult surveys in Northern Ireland [38,39]. The GHQ-12 is a screening instrument for non-psychotic psychiatric disorder and is based on answers to questions about twelve symptoms, such as concentration, sleep loss due to worry, anxiety, loss of confidence and general happiness, experienced by respondents in the period of a few weeks before the interview. For each item, there are four response categories to choose from. There are various ways to score the GHQ-12. In this study, the main outcome variable was the GHQ-12 mean score which is based on assigning each response a score of: 0 ‘not at all’; 1 ‘no more than usual’; 2 ‘rather more than usual’ and 3 ‘much more than usual’ giving a scale score ranging from 0 to 36. In order to compare GHQ-12 ‘caseness’ with previous research, the responses were coded as: 0 ‘not at all’ and ‘no more than usual’ responses, and 1 ‘rather more than usual’ and ‘much more than usual’. The item responses were summed to give a score ranging from 0 to 12. The cut-off for caseness was taken as a score of 4 or more [8,36,40]. Whilst some authors have raised concerns about the GHQ-12′s low positive predictive value and its suitability as a clinical diagnostic tool [41], nonetheless, a meta-analysis has confirmed that the GHQ-12 is essentially a unidimensional general mental health screening tool [42]. Furthermore, the GHQ-12 has been widely used in general population surveys of adults in Northern Ireland [39,43] and among 16-year-olds in YLT since 2004 [40,44]. Therefore, for comparison purposes, it was deemed the best measure of mental health for inclusion in the 2020/21 YLT survey. In this study, the Cronbach’s alpha was 0.90 demonstrating excellent internal consistency [45].

### 2.2. The COVID-19 Question Module

The COVID-19 module asked 16-year-olds to self-assess the impact of the pandemic on their physical and mental health. Questions also focused more specifically on school experiences during the lockdown including extra-curricular experiences, contact with friends and any issues they experienced in their home environment. Respondents were also asked their opinions on how they felt young people’s views were taken into consideration by government when decisions were made about COVID-19 measures, and what can be learnt from this going forward. Two open-ended questions were included in the survey:What do you think have been the most difficult things experienced by children and young people during the COVID pandemic that must be addressed by Government?What actions should Government take to make things better for children and young people as we look to move forward from the COVID pandemic period?

A large percentage of survey participants (88%) provided a response to the question on the difficult things they experienced during the pandemic, whilst 78% offered a response to the question on government measures that should be put in place as we emerge from the pandemic.

Open-ended questions have featured regularly in the YLT survey. As we argue elsewhere, open-ended questions can provide a more detailed and nuanced understanding in addition to closed questions and may raise issues that had not been considered when drafting closed survey responses [46]. As such, they potentially enrich statistical survey findings. Open-ended questions also provide ‘a degree of agency to respondents by allowing them space to voice their opinion’ [46] (p. 2). For NICCY, the funders of this COVID-19 module in the YLT survey, the option of collecting open comments was particularly attractive, as it allowed them to utilise these comments to communicate young people’s experiences of COVID-19 more poignantly to policy makers [47]. The survey questions were jointly developed with NICCY to make sure they captured what the organisation was interested in. Each year, the questions for YLT are pilot tested by a small number of young people to ensure the question wording is understood and appropriate, response burden is acceptable and valid responses are yielded.

### 2.3. Data Analysis

Before analyses were conducted, the dataset was cleaned. Respondents can skip any items they do not want to answer and these are set to missing; those who only answer a very small number of questions on YLT are removed from the dataset. As the GHQ-12 is a standardised measure, respondents who did not answer all 12 items were removed before data analyses.

Descriptive statistics (percentages, mean scores) on the characteristics of the respondents and the independent variables used in this paper are presented. The data from the GHQ-12 were analysed using multiple regression which is a statistical technique used to predict an outcome variable (GHQ-12 scores) using a number of explanatory variables. The key assumptions for linear regression—normal distribution of the residuals, absence of multicollinearity, and presence of homoscedasticity—were tested. Results indicated that there was no evidence of abnormalities in the data that would affect the regression analysis: VIF (Variance Inflation Factor) values were below 10 and tolerance values exceeded .10. The explanatory variables sex, disability, sexual attraction and socio-economic status were included in the model as previous research has demonstrated that the detrimental effects of the COVID-19 lockdowns on mental health are most notable among these groups in the population [8,10,11,12]. The effect of restrictions on contact with friends has been demonstrated to have adversely affected the mental health of young people [27]. In addition to the GHQ-12, which is a standardised measure, the young people were asked to report their own perceptions of how the COVID-19 lockdowns had affected their physical and mental health [48].

As these explanatory variables were at nominal and ordinal level, dummy variables were constructed. The reference groups were selected on the basis of normative groups (e.g., disability, same-sex attraction), previous research demonstrating better mental health among the reference group (e.g., gender, SES) or where the aim in the current study was to compare those who felt their health status had changed compared to those whose health status had stayed the same. The explanatory variables and reference groups are as follows:Gender (reference group ‘males’);Disability (reference group ‘no disability’);Same-sex attraction (reference group ‘heterosexual’);SES (reference group ‘well-off’);Whether physical heath was affected by COVID-19 (reference group ‘same’);Whether mental health was affected by COVID-19 (reference group ‘same’);Contact with friends (reference group ‘agree’).

An inductive thematic analysis was undertaken of the open questions with the aim of identifying the main issues raised by young people. In a first step, all responses to open questions were entered into NVivo 12 where, initially, word count and word search routines were performed. ‘Quantitising’ of open responses [49] is now a common approach for textual data, arguably fostered by the fact that these counting activities can be efficiently undertaken by computer-assisted qualitative data analysis (CAQDAS) software packages. In the case of the YLT study, quantitising enables the construction of quantitative variables that can be related to the other survey variables, such as GHQ-12 caseness. We acknowledge that word count and word search logarithms can only ever be a first explorative step in the analysis of free text responses. Some authors [50] have warned that the increasing use of CAQDAS software packages, such as NVivo, may result in a new orthodoxy and homogenisation in qualitative data analysis, but we simply used NVivo to efficiently organise and manage open data.

For this article, we read each open response several times, identified themes and grouped responses according to the emerging themes. We then chose quotations for inclusion in this article that best represented these main themes and their nuances. We report these here, as they provide staggering evidence for the impact that the COVID-19 pandemic had on young people’s lives, in particular, their mental health. Four main themes were identified from the open responses in the textual analysis.

## 3. Results

Table 1 presents the characteristics of the respondents to the 2020/21 YLT survey. As can be seen, there were more females than males (55% and 44%, respectively), 21% of respondents had been attracted to someone of the same sex, 16% had a long-term illness or disability and 48% thought their family financial background was ‘average’. Of note, more than half (52%) of all respondents thought their mental health had deteriorated over the lockdowns and 45% scored 4 or more on the GHQ-12 suggesting the presence of minor psychiatric disorders.

These findings indicate that COVID-19 had a negative impact on the mental health of YLT respondents and is of particular concern given the trend in deteriorating mental health that has been occurring among young people over the last few years in Northern Ireland and elsewhere [8,35]. This self-reported decline in mental health reported by the YLT respondents also echoes previous longitudinal research with young people [26].

The GHQ-12 questionnaire has been included in previous YLT surveys. Table 2 shows that GHQ-12 caseness has increased markedly since the measure was first included in YLT in 2004. At that time, around one quarter of YLT respondents (23.8–15.6% of males and 29.9% of females) scored 4 or more on the GHQ-12. Females consistently scored higher than males, as the table shows. Whilst there is a gradual increase in GHQ-12 caseness, in 2020/21 the proportion of 16-year-olds whose responses to GHQ-12 suggest the presence of minor psychiatric disorders was nearly twice as high as it was in 2004. This would suggest that over half (55.5%) of 16-year-old females, and nearly one in three (30.5%) 16-year-old males, deal with at least minor or modest mental health issues. Chi-squared analysis indicated that in 2020/21 the difference between males and females was statistically significant (χ^2^ = 122.93, df = 1, *p* < 0.001, Phi 0.25).

### 3.1. Regression Model

Hierarchical linear regression was carried out with GHQ-12 mean score as the outcome variable and the demographic variables gender, disability status, sexual attraction and family financial background entered as predictors in Model 1 (Table 3). This model was statistically significant (F = 71.56, df = 6,1876, *p* < 0.001) and the demographic variables accounted for 18% of the variation in GHQ-12 scores. Whilst all the demographic variables were statistically significant, the best predictor of poor mental health in Model 1 was gender, with females having poorer mental health compared to males. The COVID-19-related variables—physical and mental health status and contact with friends—were added in Model 2 and the model was statistically significant (F = 84.97, df = 12,1870, *p* < 0.001). Model 2 accounted for 35% of the variation in GHQ-12 mean scores. The best predictor of GHQ-12 in Model 2 was the perception that mental health had worsened over lockdown. With the addition of the COVID-19 variables, being attracted to someone of the same sex was a better predictor of poor mental health than gender.

### 3.2. Young People’s Open Responses on the COVID-19 Pandemic

#### 3.2.1. Mental Health and Lack of Mental Health Services

The words ‘mental’ (n = 653) and ‘health’ (n = 648)’ were by far the most mentioned words by YLT respondents in the question on the most difficult things experienced by young people during the pandemic. ‘Looking after young people’s mental health’ was also the main theme in the responses to the question asking what measures government should take to improve matters for children and young people as we move out of the pandemic; ‘mental health’ was mentioned 231 times in the responses to this question. This was, by far, the most mentioned substantial theme after the general demand to listen more to young people’s concerns—a point that will be addressed later.

The large number of responses that allude to mental health-related issues give a clear indication how much young people’s mental health was affected by the pandemic and the measures taken to stop the spread of the virus. Whilst most young people referred to mental health generally, some also alluded to personal specific conditions, such as eating disorders, loneliness, anxiety, depression, self-injury, suicidal ideation and stress. The severity of the issues, and young people’s frustration with the failure to put adolescent mental health support services in place, is clearly evidenced in the following quotes:

‘1. Basic mental health issues for example, I experienced suicidal thoughts and tendencies. It was proven very difficult to find any source of help. 2. There isn’t enough support for teenagers with their decreasing mental health in regards to studying and motivation.’

‘A lot of people I know have struggled with their mental health during this all and have been denied help or guidance to get through it. Personally I used the lockdown to work on myself physically and mentally and so my mental health got slightly better but is still not the best.’

#### 3.2.2. Social Interactions and Friends

It is no coincidence that ‘social interactions with friends’ was identified as the second main theme from the open responses. ‘Friends’ (n = 363) was the second most frequently mentioned issue by young people. In their responses, many 16-year-olds commented how ‘seeing friends and socialising with others is […] essential to many teenagers for their mental health and wellbeing’. One respondent stated how ‘small things like going to the cinema or shopping etc. with friends can really affect [one’s] mental health’. Another respondent reported how young people could develop ‘depression or anxiety from being separated from their friends’.

The respondents commented on how going in and out of lockdowns impacted young people’s routines, including seeing friends at weekends. One respondent stated how important this routine is for young people’s mental health:

‘Constantly going in and out of lockdowns prevented us from having a routine and seeing close friends and it affected most people’s mental health.’

#### 3.2.3. School and Exams

After ‘mental health’ and ‘friends’, ‘education’ was the third major theme identified in YLT respondents’ comments. ‘School’ (n = 341), ‘exams’ (n = 295) and ‘education’ (n = 216) were the next most frequently mentioned words in young people’s responses to the question on what issues had impacted them most during the pandemic. YLT respondents commented on the uncertainty in both delivery and assessment in education which had produced high levels of stress and anxiety and, again, impacted negatively on their mental health. The mismanagement of exams and grading was particularly pertinent for this cohort of young people, as they experienced significant disruption in their GCSE (General Certificate of Secondary Education) year. For many young people, GCSE exams are the final exams they take before leaving school and applying for jobs, apprenticeships or training programmes. GCSE results also determine whether young people can stay on in school to study for their university entry level qualifications. The chaos around GCSE exams, and subsequent cancellation of exam marks, resulted in inflated grades given by teachers compared to the exam results of pre-COVID-19 cohorts [51]. This left many 16-year-olds wondering if they were really working at the standard required to achieve university-level entrance exam results. The following responses exemplify this:

‘Education has been affected a lot, I am now going into lower sixth with no exam practice from GCSE. I am also worried about what A-levels to pick as I have missed a lot of content from GCSE. (…)’

‘How exams were carried out. It wasn’t fair that we were told that exams were cancelled one moment, then next we were told we had two weeks to learn two years’ worth of material, when we hadn’t been in school for the majority of that time. The government continues to make the same mistakes continuously in regards to the education of kids with very little regard for us (…).’

#### 3.2.4. Neglecting Young People’s Needs and Views

The last theme identified in the open responses in the YLT survey very much captures the sentiment that young people’s needs and their views were neglected in the management of the pandemic. The need to ‘listen’ to (n = 143) and ‘support’ (n = 123) young people were two of the most frequently mentioned words in the response to the question on how government should improve matters for young people, as we emerge from the pandemic. Again, respondents linked this need to the difficulties that young people experienced with regard to mental health (n = 231 in this question) and school (n = 198). This response acknowledges that COVID-19 was ‘hard on everyone’, but points out that there was little focus on children and young people’s needs:

‘Although the COVID-19 pandemic has been hard on everyone, I don’t think the government truly realises how harsh it was on us children/teenagers. These are the years where we interact and gain the most friendships and experience more things, but COVID-19 halted it rather suddenly.’

The next response also points to the perceived disregard of young people’s needs and voices in decision making—again highlighting the negative effect that the pandemic had on young people’s needs.

‘(…) Students have been disregarded in most of the decisions made and it’s apparent when you look at how stressed the average student is. The government should address why they refuse to listen to students and young people and also why they continue to strive for normality at the expense of students’ mental health.’

In their open responses, the young people demanded more accessible information and more clarity about COVID-19 regulations which were often contradictory within and between countries and regions, and they particularly wanted more clarity around exams and education in general. The responses highlighted the need to provide age-specific spaces and activities that enable young people to socialise and undertake group activities that they stressed were important to support their mental health and relieve stress. It was very evident that young people had found the lack of meeting opportunities very stressful and detrimental to their mental health, and they felt they should have been able to communicate this when decisions were made on how to manage the pandemic. Young people stated that they were not to be blamed for the pandemic, but there was a sense that they were disproportionately affected by this through the loss of an important and crucial period of growing up and development of social skills that they could not regain. Young people were also at the end of the vaccination queue, and some expressed that the prioritisation of young people in the vaccination programme could have prevented school closures.

In their responses, 16-year-olds expressed a sense of exclusion in decision making, being forgotten, and that their specific concerns were not taken seriously, which created an environment where they felt unheard and not understood. They appreciated that the pandemic had impacted them differently than older generations, but they felt that their concerns were not listened to.

‘I believe that being unable to socialise during the lockdowns has had a negative impact on many young people and children, as this is a crucial time in our lives which we would usually be developing through what we learn and do in school and with our friends, so removing this aspect from our lives for an extended period of time may result in children and young people lacking necessary social skills. Self-isolation has also been a difficult experience for many children and young people. Many students that attend my school, including myself, have had to self-isolate at some point in the past year, with some having to self-isolate for even longer. Self-isolating was definitely detrimental to the mental health of young people. Being stuck inside for two weeks, unable to see your friends and only seeing your family from a distance or through a screen was difficult to cope with, and I personally underestimated how difficult I would find it.’

## 4. Discussion

The data from the 2020/21 YLT survey show that mental health among 16-year-olds in Northern Ireland, as assessed by the GHQ-12, is poorer now than in previous years. There has been a marked rise in the percentage of ‘cases’—young people scoring 4 or more on the GHQ-12—suggesting the presence of mental health issues. This aligns with national and international evidence that has reported a declining trend in mental health and wellbeing among young people [35,38]. Whilst inconclusive, some research does suggest that potential reasons for this decline in mental health include the negative effects of social media (e.g., cyber-bullying, fear of missing out, body image dissatisfaction) [52,53] and school pressures (e.g., increasing use of high-stakes testing in the UK and elsewhere) [54,55].

Furthermore, the results from the regression analysis have clearly shown the negative impact that COVID-19 has had on the mental health of young people in Northern Ireland. These findings support previous research from studies conducted in other regions of the UK and internationally [5,8,12,13]. Research has demonstrated that isolation during COVID-19 lockdowns led to increased anxiety and stress [56]. Additionally, the enforced move to online learning and the cancellation of high-stakes examinations has led to an increase in levels of academic stress and feelings of pressure to make up for the loss of learning experienced during the lockdowns [57,58,59].

Of note, the results from the regression analysis in Model 1 showed that female participants experienced a greater decline in mental health during COVID-19 than their male counterparts, supporting previous research [11,28]. However, whilst gender remained statistically significant in Model 2, the addition of the COVID-19-related variables (physical and mental health status and contact with friends) and identifying as non-heterosexual were more important in predicting poor mental health than gender. The finding that young people who identify as non-heterosexual have poorer mental health than their heterosexual peers offers support for previous research which found poorer mental health among this vulnerable group both pre- and post-pandemic [28,29,30,31]. One of the key findings emerging from Model 2 was that young people who felt that their mental health had worsened over lockdown had lower GHQ-12 scores than their peers who thought their mental health had stayed the same or improved. This suggests a good level of awareness of mental health issues among young people, something that has been noted in other research [52,60,61]. Evidence for this comes from the results of systematic reviews which have shown that young people use social media to seek information on mental health issues and explore ways to help alleviate their symptoms [52,61].

The findings from the statistical analyses were supported by the wealth of information provided by the young people in their open-ended responses. These young people identified several factors that they believed impacted their mental health during COVID-19, such as a lack of emphasis on, and availability of, mental health services provision specifically targeting adolescents, due to long waiting times to access services, limited number of practitioners providing direct services, and inability to receive services at a convenient location due to the lockdown. These findings are well supported by empirical evidence, with the World Health Organisation (WHO) reporting that the pandemic has disrupted or halted critical mental health services in 93% of countries worldwide [62]. Similarly, research commissioned by NICCY found that access to mental health care services for children and young people in Northern Ireland worsened during COVID-19 [47].

The open responses from young people collected in the YLT survey also support the UK All-Parliamentary Group’s sentiment that we are faced with a ‘mental health pandemic among young people—a disaster ‘– as a fallout of COVID-19 [63]. When asked what the main issues were that young people experienced during the pandemic, and what should be addressed by government, YLT respondents wrote about the social isolation and loneliness, the lack of opportunities to socialise with friends and peers, their anxiety, the stress they experienced about exams, and the resultant uncertainty about their future. A large proportion of YLT respondents expressed clearly that the government needed to put better support services in place for young people. All of these issues related to the pandemic may go a long way to explain the significant increase in GHQ-12 caseness recorded in the 2020/21 YLT survey compared to previous years.

A lack of social interactions with friends and those outside of one’s household were noted by participants to be a major source of distress during the lockdowns. Young people reported at length that disruption of their normal social routines, caused by going in to and out of multiple rounds of lockdowns, resulted in significant upheaval in their social and emotional lives. Routines and frequent peer-to-peer social interactions have been shown to create networks of support that serve as protective mechanisms in the developing neural pathways of the prefrontal cortex, the portion of the brain responsible for coordinating executive functions such as complex social interactions, self-regulation, and emotional maturity [64]. In young people, school attendance and extracurricular activities provide the primary basis for such patterning and functioning. With the disruption of these routines caused by the series of lockdowns, the emotional distress expressed by the participants is a logical sequela of these events [65]. Returning to routine social engagements and school activities can improve mental health in young people [66]. However, it is yet to be determined if the prolonged disruption of such routines, secondary to the COVID-19 lockdowns, will have long lasting impacts on the mental health of these young people as they mature.

School stressors were also highlighted in this study. Participants noted that the disruption in their schooling was a major factor impacting their mental health from a social isolation standpoint. Additionally, students noted that concerns over their formal examinations were a major source of stress that contributed to poor mental health overall. Arguably, the uncertainty about educational qualifications and the mismanagement of assessments and exams in particular led to increased levels of anxiety among young people about their future [67]. Whilst as we acknowledge above, a quantification of responses can at best be a first step in an attempt to make sense of qualitative data, the sheer volume of responses received in YLT on these core issues made the identification of the four main themes a relatively straightforward task.

Lastly, the participants felt that young people were not prioritised by the government’s response efforts and their needs remained unmet. A major source of the disgruntlement reported in this study was the lack of communication regarding the decisions that were made that directly affected young people, and the lack of input they were given in the decision-making process resonating with other research findings [33,68]. Perception of control, over one’s decisions, daily activities, future plans, etc., is not only a contributing factor to the maturation of young people, but they also have a right to be consulted about decisions that affect them, and COVID-19 lockdowns certainly did. When decisions are made outside of a young person’s perceived locus of control, significant emotional distress may occur [69].

## 5. Limitations and Strengths

A limitation of this research is that it relied on cross-sectional data that cannot be used to provide causal evidence of the direct effect of COVID-19 lockdowns on mental health. Nonetheless, the findings offer support for previous longitudinal research from other regions of the UK and countries across the globe. Secondly, there is an over-representation of females (55%) compared to males (44%), perhaps skewing the results somewhat. Thirdly, the study used the GHQ-12, a self-report measure which research has shown is more likely to lead to under- rather than over-reporting of mental health issues due to the stigma associated with mental illness [70,71]. A more open debate of mental health in society may reduce this stigma and may, over time, lead to an increase in the proportion of young people recognising and reporting mental health issues. Finally, as with all research, there is a possible inherent response bias as those who feel strongly about the research topic may be more inclined to consent to take part, and indeed may be more willing to provide more detailed open responses.

Despite these limitations, the study has a number of strengths. Firstly, the quantitative analysis provided data on the predictors of mental health among a large sample of young people aged 16 years—an age at which the onset of all mental health disorders is at their peak [72]. Secondly, the survey included open-ended questions that provided rich data from the high percentage of young people who took the time to answer. The open responses align well with the quantitative data, and also with the results of other studies which have all shown a substantially negative impact of COVID-19 measures on young people. Thirdly, the study was timely with the fieldwork being conducted when the UK was experiencing a cycle of lockdowns and when young people were facing much uncertainty about their examination assessments and the implications this could have on their future prospects.

## 6. Conclusions

As demonstrated by this research, young people identified that the lack of control over their daily lives, the lack of consistent peer social interaction, and a feeling that they were excluded from decision-making processes greatly impacted their mental health. These concerns are further confirmed by the GHQ-12 scores, demonstrating a global decline in mental health for this population. This will add to the pressures on the already stretched and under-sourced mental health services. It is clear that further investment in adolescent mental health services will be required to meet the needs of those presenting with mental health issues, but, going forward, the learning from the pandemic should also be that we need to improve our understanding of protective factors that foster positive mental health among young people.

Some studies have shown that engaging young people in a decision-making capacity can help to improve their emotional response to traumatic and stressful situations [73]. Moving forward, as governments and institutions prepare for future pandemics, crises, and disruptions, efforts should be made to ensure that young people are included in the decision-making processes and that responses to their concerns are clearly communicated, as a way to help mitigate against negative mental health outcomes.

To conclude, COVID-19 has only just begun to recede and the impacts of it, and its associated lockdowns, have yet to be fully described and understood. As we continue to move forward, attention must be paid to not only the physical health, but equally the mental health, of those young people who experienced the pandemic first-hand. Early studies such as this one demonstrate the negative consequences of COVID-19 and illuminate how governmental policies have affected the mental health of young people. With this level of disruption experienced in their formative years, the impact of these experiences may well continue for the remainder of their lives. As with any traumatic event, be it formal wars, civil unrest, disasters, or even health crises, attention and efforts must be put in place by governments, not-for-profit organisations, the medical profession, the mental health community, and academia to ensure young people can recover from their experiences and return to full function within society as they continue to mature into adulthood.

## Figures and Tables

**Table 1 ijerph-20-06550-t001:** Characteristics of young people taking part in YLT 2020/21.

	n (%)	Mean (SD)
*Gender*		
Male	870 (44)	
Female	1100 (55)	
Other	25 (1)	
*Sexual Attraction*		
Opposite sex attracted	1436 (74)	
Same sex attracted	402 (21)	
Never sex-attracted	104 (5)	
*Disability*		
Yes	320 (16)	
No	1664 (84)	
*Perception of family financial background*		
Well-off	659 (33)	
Average	958 (48)	
Not well-off	226 (11)	
Don’t know	149 (8)	
*Mental health during lockdown*		
Worse	1035 (52)	
Better	265 (13)	
The same	681 (34)	
*Contact with friends during lockdown*		
Agree	1656 (83)	
Neither	136 (7)	
Disagree	201 (10)	
*GHQ-12 Caseness (Score 4+)*		
Not a case	1098 (55)	
Case	897 (45)	
GHQ-12 (0–36)		14.33 (7.31)

**Table 2 ijerph-20-06550-t002:** YLT GHQ-12 scores over the years–caseness method (score of 4 or more) ^1^.

Year	All (%)	Males (%)	Females (%)
2004	23.8	15.6	29.9
2005	21.4	11.5	28.0
2006	19.8	14.9	23.3
2007	21.0	15.0	25.2
2008	28.6	18.3	35.3
2011	28.2	19.2	35.6
2013	29.3	18.5	37.4
2020/21	45.0	30.5	55.5

^1^ *p* < 0.001; effect size 0.25 (Phi), only cases included with all 12 questions answered.

**Table 3 ijerph-20-06550-t003:** Regression model GHQ-12.

	Model 1		Model 2	
Variable	B *(se)*	β	B *(se)*	β
Constant	10.61 *(.30)*	.297	7.781 *(.329)*	.329
*Gender*				
Male (Ref)	-	-	-	-
Female	3.74 *(.31)* ***	.255	2.50 *(.29)* ***	.171
*Sexual Attraction*				
Opposite sex attracted (Ref)	-	-	-	-
Same sex attracted	4.16 *(.39)* ***	.230	3.33 *(.35)* ***	.184
Never sex-attracted	−.535 *(−.680)*	−.017	−.23 *(.61)*	−.007
*Disability*				
No (Ref)	-	-	-	-
Yes	2.75 *(.427)* ***	.136	2.03 *(.38)* ***	.100
*Perception of family financial background*				
Well-off (Ref)	-	-	-	-
Average	.162 *(.323)*	.011	.083 *(.29)*	.006
Not well-off	2.487 *(.515)* ***	.108	1.42 *(.46)* **	.062
*Physical health during lockdown*				
The same (Ref)	-	-	-	-
Worse	-	-	1.415 (.32) ***	.090
Better	-	-	.002 (.35)	.000
*Mental health during lockdown*				
The same (Ref)	-	-	-	-
Worse	-	-	5.55 *(.322)* ***	.382
Better	-	-	1.45 *(.440)* ***	.068
*Contact with friends during lockdown*				
Agree (Ref)	-	-	-	-
Neither	-	-	2.21 *(.546)* ***	.076
Disagree	-	-	2.93 *(.458)* ***	.122
*Adjusted r square*	.184		.349	

** *p* < 0.01; *** *p* < 0.001.

## Data Availability

Data collected for this study are freely available from www.ark.ac.uk/ylt/datasets (accessed on 18 February 2023).

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
