# Peer review of "A Mental Health Pandemic? Assessing the Impact of COVID-19 on Young People’s Mental Health"

_ijerph, 2023, doi:10.3390/ijerph20166550_

Round 1
Reviewer 1 Report
OVERALL. This is a relevant text for a broad, cross-field readership for the following main reasons: i. Topicality of the subject matter; ii. Focus on young people, thus the population that has been shown to be the most fragile in the post-acute period of the Covid-19 pandemic); iii. Clear practical implications (so important to the clinical reader).
IN DETAIL.
- Discussion. It can be enriched and made more original, precisely for the purpose of bringing out even more prominently the relevance of the study in current peer-review. Two points I invite the Authors to mention: 1) In my opinion, it is imperative to mention among the study's assumptions in this context the fears that this population had in relation to the Covid-19 pandemic. I suggest that the Authors consult and use the following paper, which has a twofold advantage: it both analyzes the specific fears of this population group and does so from the pragmatic perspective of prevention/rehabilitation (thus both aspects in perfect alignment with the Authors' text). This is a study conducted at Psychiatric Emergencies, a front-line service that constitutes a privileged observatory as it often anticipates what will be the experiences in the general population. It is: "COVID-19 Related Fears of Patients Admitted to a Psychiatric Emergency Department during and Post-Lockdown in Switzerland: Preliminary Findings to Look Ahead for Tailored Preventive Mental Health Strategies. Medicine (Kaunas). 2021 Dec 13;57(12):1360. doi: 10.3390/medicina57121360"; 2) The fact that gender differences in having suffered in relation to this pandemic have emerged, and that this is also attributable to the quality of the environment in which the young people spent the lockdown: "Gender Differences in COVID-19 Lockdown Impact on Mental Health of Undergraduate Students." Front Psychiatry. 2022 Jan 5;12:813130. doi: 10.3389/fpsyt.2021.813130.
Best regards.
Minor editing
Author Response
Dear reviewer of the International Journal of Environmental Research and Public Health,
We would like to acknowledge your kind peer review of our article ‘A Mental Health Pandemic? Assessing the Impact of COVID-19 on Young People’s Mental Health’.
We have now reviewed and revised this article. Thank you for your patience with us during the time we made our revisions. We received 5 very diverse peer reviews, and we attempted to take on board the comments from all reviewers, but these were sometimes contradictory.
In a nutshell:
- We have revised and restructured our abstract to reflect the expected structure and the amendments in our article.
- We have given our introduction a more logical order by themes. We removed material not directly related to our own data to make space for some of the other requested additions. The introduction now focuses on literature that also discusses the mental health of young people pre- and post-COVID-19, in particular the difference between LGBTQ+ young people and heterosexual cisgender young people, the difference between males and females and finally specific trends in the UK and Northern Ireland where our own study was situated.
- We have added literature suggestions made by all reviewers if they fell under these themes if we had overlooked these in our own background research. Thank you for these suggestions.
- We provided more specific details on our survey methodology, both the GHQ-12 instrument and why we used this to assess young people’s mental health, and the actual YLT survey, including methodological limitations. Here, we specifically provided more details on why the YLT survey is a suitable tool and why its sampling frame was the most suitable one for our study; we addressed the issue of sampling bias and dataset cleaning. We also provided more detail on how we used the qualitative data and how the open responses were coded, thematically organised and then chosen for inclusion.
- The peer reviews were contradictory with regard to the need for an inclusion of a hypothesis and whether or not we ought to include policy recommendations. We discussed this and decided not to retro-fit a hypothesis, and therefore also largely maintain the structure of the conclusion. We did, as requested by some reviewers, strengthen the policy recommendations, in particular in relation to the need to include young people more in policy decisions and to improve mental health service provision for young people.
Specifically, we have taken on board your suggestions to reference the article on gender differences - thank you for this recommendation. We studied the other recommended article too, but on reflection, we decided against inclusion, as our study population was one randomly drawn among 16-year olds in the general population, and we felt that the issues experienced by those in psychiatric wards was substantially different. 'Fear' did not come up as a theme in the open questions in our survey, and we felt we ought to reflect our data.
Many thanks again for your valuable comments.
Reviewer 2 Report
A Mental Health Pandemic? Assessing the Impact of COVID- on Young People’s Mental Health
Overall, the paper presents an important and timely analysis of the impact of the COVID-19 pandemic on the mental health of 16-year-olds in Northern Ireland. It highlights the worsening mental health trends among young people and identifies factors contributing to these issues, such as social isolation, disrupted routines, and feelings of exclusion from decision-making processes.
Strengths of the paper include:
- The use of both quantitative and qualitative data, provides a comprehensive understanding of the mental health issues faced by young people during the pandemic.
- The large sample size, which allows for greater generalizability of the results.
- The focus on a specific age group (16-year-olds) that is particularly vulnerable to mental health issues and has experienced significant disruptions to their education and social lives due to the pandemic.
However, there are some areas for improvement:
- Addressing the limitations of the study, including the reliance on cross-sectional data, self-reporting biases, and potential under-representation of certain demographics.
- Providing more concrete recommendations and strategies for governments, institutions, and other stakeholders to address the mental health needs of young people during and after the pandemic.
- Discussing the importance of continued research and monitoring of mental health trends among young people, even as the pandemic recedes, to ensure that emerging issues are identified and addressed promptly.
By addressing these points and incorporating the suggested improvements, the paper will make a valuable contribution to the existing literature on the impact of the COVID-19 pandemic on young people's mental health and provide important insights for policymakers and practitioners in addressing these issues.
in particular:
The abstract could be improved in the following ways:
- Structure: The abstract can benefit from a more organized structure, presenting the background, objectives, methods, results, and conclusions in a sequential and coherent manner.
- Conciseness: Some sentences can be made more concise, providing a clearer and more direct message.
- Connection between results and implications: The abstract should better explain how the results lead to the conclusion that governments need to listen to and put better support services in place for young people.
- Future research: The abstract could briefly mention potential areas for further investigation or any limitations of the study that future research should address.
The introduction could be improved in the following ways:
- Structure: Organize the content by presenting the key points in a more logical and coherent manner, focusing first on the gender differences, then discussing the mental health of non-heterosexual individuals, and finally addressing the mental health trends in the UK and Northern Ireland.
- Flow and coherence: Ensure that each paragraph is focused on a specific topic, and use transitions to smoothly connect one paragraph to the next.
- Conciseness: Some sentences can be shortened or restructured to provide a more direct and focused message.
- Balance: Make sure that equal attention is given to each key point, avoiding disproportionate focus on one aspect over another.
- Context: Provide context and background for the studies mentioned, explaining why they are relevant to the current research.
- Terminology: Define or explain any specific terms or concepts related to mental health, COVID-19, or the studies discussed, ensuring that readers from various backgrounds can understand the content.
- Clear objectives: Clearly state the objectives of the study and the research questions that the paper aims to answer, ensuring that the reader understands the purpose of the study.
- Connection to the rest of the paper: Ensure that this section sets the stage for the subsequent sections so that readers can easily follow the flow of the paper.
Here are some suggestions to improve the "3. Methods" section:
- Be more explicit about the rationale for using the YLT survey for this study. Explain why the YLT survey is suitable for collecting data on mental health and COVID-19 experiences among 16-year-olds in Northern Ireland.
- Consider clarifying the use of the Child Benefit Register as a sampling frame. Explain how the sampling frame ensures representativeness and why it is suitable for this study.
- Provide more details on the data cleaning process. Explain the criteria for removing respondents and any data quality checks performed before analysis.
- In the GHQ-12 subsection, explain the rationale for choosing this instrument for mental health assessment. Also, consider discussing any limitations of using the GHQ-12, such as cultural differences or possible response bias.
- The COVID-19 question module subsection provides more information on how the questions were developed and validated. Describe the process of question selection and any pretesting done to ensure their reliability and validity.
- Discuss the representativeness of the respondents who answered the open-ended questions. Address whether their perspectives might be different from those who did not provide responses.
- Address potential limitations in the study design, such as potential biases or challenges in interpreting the data. This could include discussing nonresponse bias or limitations in generalizability.
Here are some suggestions to improve the "3.3. Data analysis" section:
- Consider discussing the assumptions and prerequisites for using multiple regression analysis. Briefly explain how these assumptions were met or addressed in the study.
- Explain the rationale behind selecting the specific explanatory variables for the regression analysis. Address why these variables are relevant and important for understanding the relationship between mental health and COVID-19 experiences.
- Clarify the coding process for creating dummy variables. Provide more detail on how the categories were determined and how the reference groups were chosen.
- When discussing the inductive thematic analysis, provide more information on the steps to ensure rigour and reliability. For example, describe how themes were identified, refined, and validated during analysis.
- Address potential limitations of using NVivo 12 and quantifying open responses. Discuss the balance between maintaining the richness of qualitative data and creating quantitative variables for analysis.
- In the discussion of the four main themes identified from the open responses, provide examples of representative quotes or statements to illustrate each theme.
Here are some suggestions to improve the "4. Results" section:
- Consider providing a more detailed description of the results in Table 1. Discuss the key findings and their implications, such as the high percentage of respondents who reported a decline in mental health during the lockdowns and the proportion with potential minor psychiatric disorders.
Here are some comments to improve the discussion section:
- Consider reorganizing the section to provide a clearer flow of information. Start by presenting the main findings and then delve into each specific result and its implications. For example, begin by discussing the overall decline in mental health among 16-year-olds and then explore the specific factors that have contributed to this decline.
- When discussing the regression analysis results, consider using more straightforward language to explain the findings. For instance, instead of saying, "gender was the best predictor of poor mental health," you could say “Female participants had a greater decline in mental health compared to male participants." This will make the findings more accessible to a wider audience.
- When discussing the impact of COVID-19 on mental health, consider providing more context on how the pandemic has specifically affected young people in Northern Ireland. This could include data on school closures, economic hardships, and social restrictions, which would help illustrate this population’s unique challenges.
- In discussing the open-ended responses, providing direct quotes from participants to illustrate their experiences and perspectives would be helpful. This would give the reader a better understanding of the issues and concerns the young people raised.
- As you highlight the importance of social interactions, school stressors, and the government’s perceived lack of prioritization of young people, consider discussing potential interventions or policy changes that could address these issues. This could involve suggestions for improving mental health services, increasing support for educational institutions, and promoting youth participation in decision-making processes.
Here are some suggestions for improvement in the Limitations and strengths
Consider mentioning the possible limitation of self-reporting bias, as the study relies on participants' self-reported mental health assessments. This might result in under-reporting or over-reporting of mental health issues. The limitation is that the sample may not be fully representative of the entire population of 16-year-olds in Northern Ireland, as it may exclude certain demographics or groups who did not participate in the study.
Here are some suggestions for improvement in the Conclusions:
- Strengthen the conclusion by emphasizing the importance of addressing the mental health needs of young people and providing appropriate support and services.
- Discuss the potential role of community-based interventions, schools, and other stakeholders in promoting mental health and well-being among young people.
- Consider highlighting the importance of early intervention and prevention strategies in mitigating the long-term impact of the pandemic on young people's mental health.
- You might also emphasize the need for ongoing monitoring of mental health trends among young people, even as the pandemic recedes, in order to ensure that any emerging issues are identified and addressed promptly.
- Lastly, reiterate the importance of including young people in decision-making processes and ensuring that their voices are heard, as a key strategy for promoting their mental health and well-being.
The manuscript titled "The Impact of the COVID-19 Pandemic on the Mental Health of 16-Year-Olds in Northern Ireland" is well-written overall, and the language used effectively communicates the research findings. However, there are some areas where improvements can be made to enhance the clarity and readability of the paper.
- Some sentences are overly long and complex, making them difficult to understand. Breaking them down into shorter sentences would improve readability and clarity.
- The manuscript contains occasional grammatical errors, such as subject-verb agreement issues and incorrect use of articles. These should be corrected to enhance the overall quality of the paper.
Author Response
Dear reviewer of the International Journal of Environmental Research and Public Health,
We would like to acknowledge your kind peer review of our article ‘A Mental Health Pandemic? Assessing the Impact of COVID-19 on Young People’s Mental Health’.
We have now reviewed and revised this article. Thank you for your patience with us during the time we made our revisions. We received 5 very diverse peer reviews, and we attempted to take on board the comments from all reviewers, but these were sometimes contradictory. Your review was by a very long way the most comprehensive one, so we started off by addressing the issues you raised and then checked any outstanding issues form the other reviews.
In a nutshell:
We have revised and restructured our abstract to reflect the expected structure and the amendments in our article.
We have given our introduction a more logical order by themes. We removed material not directly related to our own data to make space for some of the other requested additions. The introduction now focuses on literature that also discusses the mental health of young people pre- and post-COVID-19, in particular the difference between LGBTQ+ young people and heterosexual cisgender young people, the difference between males and females and finally specific trends in the UK and Northern Ireland where our own study was situated.
We have added literature suggestions made by all reviewers if they fell under these themes if we had overlooked these in our own background research. Thank you for these suggestions.
We provided more specific details on our survey methodology, both the GHQ-12 instrument and why we used this to assess young people’s mental health, and the actual YLT survey, including methodological limitations. Here, we specifically provided more details on why the YLT survey is a suitable tool and why its sampling frame was the most suitable one for our study; we addressed the issue of sampling bias and dataset cleaning. We also provided more detail on how we used the qualitative data and how the open responses were coded, thematically organised and then chosen for inclusion.
The peer reviews were contradictory with regard to the need for an inclusion of a hypothesis and whether or not we ought to include policy recommendations. We discussed this and decided not to retro-fit a hypothesis, and therefore also largely maintain the structure of the conclusion. We did, as requested by some reviewers, strengthen the policy recommendations, in particular in relation to the need to include young people more in policy decisions and to improve mental health service provision for young people.
We proof-read the article several times.
Many thanks again for your valuable comments.
Reviewer 3 Report
Dear Authors,
I read with interest your article which highlights the possible mental health problems during the Covid-19 pandemic in your country, as well as anywhere in the world.
There are several aspects to which I can draw your attention:
1. Abstract: it must be structured in Background, Materials and Method, Results and Conclusion , in approximately 200 words.
2. Introduction: First phrase with numbers of death up to date is not relevant for this article, maybe you should find something related to the period you studied. The exact date of the study is not specified, 2020/2021 is quite ambiguous, it appears from the article that data processing started in May 2021
Material and Method: It is quite difficult to understand the whole process, you offered a lot of explanations that are not really related to this study. For ex. there are a lot of information about Child Benefit social system explanation that are not related to this article. Please, remove.
I understood that each study participant received a 10 lira voucher. Is this mentioned in the ethics agreement received for this study? As a rule, study participants are volunteers, they are not remunerated for their participation.
Please rephrase line 202-203-205, line 236 (physical health), line 313 (quote number 3?), In general rephrasing the main idea of each item of open answer, without quoting-it would be much easier to follow the most important aspect, in my opinion, it is hard to follow the rest if you are underlying many times the same idea. (line 374-384; 405-415, etc.)
Discussions: I recommend short phrases without repetition of results that were already presented on "Results".
And one more aspect, how can you explain the small difference of 99 persons between 1035 young people that experienced worse mental health and the rest of 265 (better)+ 681 (the same)?
Success!
Author Response
Dear reviewer of the International Journal of Environmental Research and Public Health,
We would like to acknowledge your kind peer review of our article ‘A Mental Health Pandemic? Assessing the Impact of COVID-19 on Young People’s Mental Health’.
We have now reviewed and revised this article. Thank you for your patience with us during the time we made our revisions. We received 5 very diverse peer reviews, and we attempted to take on board the comments from all reviewers, but these were sometimes contradictory.
In a nutshell:
We have revised and restructured our abstract, as requested by you and another reviewer, to reflect the expected structure and the amendments in our article.
We have given our introduction a more logical order by themes. We removed material not directly related to our own data to make space for some of the other requested additions. The introduction now focuses on literature that also discusses the mental health of young people pre- and post-COVID-19, in particular the difference between LGBTQ+ young people and heterosexual cisgender young people, the difference between males and females and finally specific trends in the UK and Northern Ireland where our own study was situated.
We have added literature suggestions made by all reviewers if they fell under these themes if we had overlooked these in our own background research. Thank you for these suggestions.
We provided more specific details on our survey methodology, both the GHQ-12 instrument and why we used this to assess young people’s mental health, and the actual YLT survey, including methodological limitations. Here, we specifically provided more details on why the YLT survey is a suitable tool and why its sampling frame was the most suitable one for our study; we addressed the issue of sampling bias and dataset cleaning. We also provided more detail on how we used the qualitative data and how the open responses were coded, thematically organised and then chosen for inclusion.
The peer reviews were contradictory with regard to the need for an inclusion of a hypothesis and whether or not we ought to include policy recommendations. We discussed this and decided not to retro-fit a hypothesis, and therefore also largely maintain the structure of the conclusion. We did, as requested by some reviewers, strengthen the policy recommendations, in particular in relation to the need to include young people more in policy decisions and to improve mental health service provision for young people.
Specifically, in relation to your comments, in addition to the restructuring of the abstract, we also revised the section on child benefit payments. This was difficult, as another reviewer actually required MORE and not LESS information. They asked us to specify why the Child Benefit Register provided the best sampling frame for our study, and this requires some detail. Hopefully, this is revised now in a way that it does contain just the right amount of information - not too much, but enough for an audit trail.
Each participant received a 10 Pound gift voucher as an incentive, yes. This is a very common method of encouraging people to participate in research , and some of the most well-known and long-running surveys use incentives like this. This does not mean that informed consent is in any way impacted. Nobody is coerced to take part. Incentives have the advantage that they encourage hard-to-reach groups more to participate, so, if used well, they have a very positive effect on response rate and reduce response bias. Ethical approval to use incentives is in place.
We have revised and proof-read the whole article.
On your last point, some respondents did not answer every question, so there are some missing values.
Many thanks again for your valuable comments.
Reviewer 4 Report
Summary of the study: The study targeted young people, who have been particularly vulnerable to negative mental health outcomes during the COVID-19 pandemic. This study used data from a survey conducted in Northern Ireland to investigate the factors related to poor mental health among 16-year-olds during lockdown. A total of 2,069 16-year-olds participated in the survey, and their mental health was assessed using the 12-item General Health Questionnaire (GHQ-12). Regression analysis revealed that being female, identifying as non-heterosexual, and perceiving a worsening of mental health during lockdown were the strongest predictors of poor mental health. Open responses supported the quantitative findings, indicating that young people felt vulnerable in terms of their mental health and educational outcomes, and believed that their needs were not prioritized during lockdown. The study's results align with findings from other countries, highlighting the adverse impact of COVID-19 on the mental health of young people in Northern Ireland. Respondents emphasized the importance of governments listening to their concerns and implementing better support services for young people in the event of future catastrophic events.
In terms of statistics, descriptive statistics were used to present the characteristics of the respondents and the independent variables. The GHQ-12 data were analyzed using multiple regression, a statistical technique that predicts the outcome variable (GHQ-12 scores) based on several explanatory variables. The explanatory variables in this study were at the ordinal level, so dummy variables were created to account for this. The explanatory variables included gender, disability, same-sex attraction, socioeconomic status, the impact of physical and mental health due to COVID-19, and contact with friends. Additionally, inductive thematic analysis was conducted.
The findings of this research indicate that young people experienced significant negative effects on their mental health due to factors like a lack of control over their daily lives, limited social interaction with peers, and feeling excluded from decision-making processes. These concerns were supported by the GHQ-12 scores, which demonstrated a global decline in mental health among young people.
Comment # 1 (minor): The conclusion of the study is a response to the hypothesis of the study and does not include recommendations. Please modify.
Comment #2 (minor): I would rate the manuscript's English level as 7 out of 10. The information is clearly presented, and the findings are supported by relevant evidence. However, there are a few areas where the writing could be improved for better clarity and flow. Some sentences could be rephrased to enhance readability, and it would be beneficial to separate the information into smaller paragraphs for easier digestion. Here is the re-writing part of the manuscript.
“The data from the 2020/21 YLT survey indicates that the mental health of 16-year-olds in Northern Ireland, as assessed by the GHQ-12, is lower now than in previous years. There has been a notable increase in the percentage of young people scoring 4 or more on the GHQ-12 - suggesting the presence of mental health issues. This aligns with national and international evidence that has reported a declining trend in mental health and well-being among young people [36, 51, 52].
The results from the regression analysis and responses to open-ended questions clearly demonstrate the detrimental impact of the COVID-19 pandemic on the mental health of young people in Northern Ireland. These findings support previous research conducted in other regions of the UK and internationally [5, 8, 13, 14].
Regarding the predictors of poor mental health, the regression analysis showed that gender was the strongest predictor in Model 1, supporting previous studies that found females experienced a greater decline in mental health during COVID-19 compared to males [12, 28]. However, while gender remained statistically significant in Model 2, the addition of COVID-19-related variables (physical and mental health status and contact with friends) revealed that identifying as non-heterosexual was a better predictor of poor mental health than gender. This finding aligns with previous research indicating poorer mental health among non-heterosexual individuals, both before and after the pandemic [11, 29, 30, 31]. The perception that mental health had worsened during lockdown emerged as the strongest predictor of GHQ-12 scores in Model 2, indicating a good level of awareness of mental health issues among young people.
The statistical analysis findings were supported by the wealth of information provided by the young people in their open-ended responses. These young people identified several factors that they believed affected their mental health during COVID-19, such as a lack of emphasis on and availability of mental health services specifically targeting adolescents. They mentioned long wait times to access services, a limited number of practitioners providing direct services, and difficulties in receiving services at convenient locations due to the lockdown. These findings are in line with empirical evidence, with the World Health Organization (WHO) reporting that the COVID-19 pandemic has disrupted or halted critical mental health services in 93% of countries worldwide [53]. Similarly, research commissioned by NICCY found that access to mental health care services for children and young people in Northern Ireland worsened during Covid-19 [48].”
This reviewer rates the English level 7/10. Please refer to the comments.
Author Response
Dear reviewer of the International Journal of Environmental Research and Public Health,
We would like to acknowledge your kind peer review of our article ‘A Mental Health Pandemic? Assessing the Impact of COVID-19 on Young People’s Mental Health’.
We have now reviewed and revised this article. Thank you for your patience with us during the time we made our revisions. We received 5 very diverse peer reviews, and we attempted to take on board the comments from all reviewers, but these were sometimes contradictory.
In a nutshell:
We have revised and restructured our abstract to reflect the expected structure and the amendments in our article.
We have given our introduction a more logical order by themes. We removed material not directly related to our own data to make space for some of the other requested additions. The introduction now focuses on literature that also discusses the mental health of young people pre- and post-COVID-19, in particular the difference between LGBTQ+ young people and heterosexual cisgender young people, the difference between males and females and finally specific trends in the UK and Northern Ireland where our own study was situated.
We have added literature suggestions made by all reviewers if they fell under these themes if we had overlooked these in our own background research. Thank you for these suggestions.
We provided more specific details on our survey methodology, both the GHQ-12 instrument and why we used this to assess young people’s mental health, and the actual YLT survey, including methodological limitations. Here, we specifically provided more details on why the YLT survey is a suitable tool and why its sampling frame was the most suitable one for our study; we addressed the issue of sampling bias and dataset cleaning. We also provided more detail on how we used the qualitative data and how the open responses were coded, thematically organised and then chosen for inclusion.
The peer reviews were contradictory with regard to the need for an inclusion of a hypothesis and whether or not we ought to include policy recommendations. We discussed this and decided not to retro-fit a hypothesis, and therefore also largely maintain the structure of the conclusion. We did, as requested by some reviewers, strengthen the policy recommendations, in particular in relation to the need to include young people more in policy decisions and to improve mental health service provision for young people.
We hope that our restructuring addresses your comment on the need to be clearer and to separate these out more. We have generally attempted to re-write the article more clearly in line with your recommendations.
Many thanks again for your valuable comments.
Reviewer 5 Report
In the current paper, the authors aimed to investigate the effects of the lockdown during COVID-19 on the mental health of 16-years old in Northern Ireland. Poor mental health was predicted by the female gender, identifying as non-heterosexual and perceiving that mental health had worsened during the lockdown. In addition, young people expressed warning about their mental health and education.
The work is interesting and well-written, and the sample is convenient for this study design. However, I have some minor concerns the authors should consider before publication.
1) My major concern regards the use of open-ended questions in the Young Life and Times survey, which has been reported in previous publications but needs further discussion. In particular, the author did not mention any limitation in using open-ended questions, especially since the results are difficult to compare and noise or irrelevant information may emerge. Expanding the discussion in the limitation section may be important.
2) Authors should check for typos and abbreviation: some abbreviation are not needed, for example PTSD and CCR, which are cited once, and other terms should be abbreviated at their first appearance, for example GHQ-12.
3) Please provide reference for the sentence “The reporting of post-traumatic stress disorders also increased”
4) I suggest the authors to add some recent or important papers which may improve the reference list:
- DOI: 10.1371/journal.pone.0279963 and DOI: 10.3390/ijerph192315795, regarding higher neuropsychiatric symptoms related to pandemic in sexual minority (discussion, page 11)
- DOI: 10.3390/jcm10215169, regarding the negative effects of COVID-19 pandemic on mental health and the higher levels of depression, anxiety and neuropsychiatric symptoms (introduction, page 1)
- DOI: 10.3389/fpsyg.2021.746289, about the effects of school closure on student achievements (discussion, page 12)
English requires minor check
Author Response
Dear reviewer of the International Journal of Environmental Research and Public Health,
We would like to acknowledge your kind peer review of our article ‘A Mental Health Pandemic? Assessing the Impact of COVID-19 on Young People’s Mental Health’.
We have now reviewed and revised this article. Thank you for your patience with us during the time we made our revisions. We received 5 very diverse peer reviews, and we attempted to take on board the comments from all reviewers, but these were sometimes contradictory.
In a nutshell:
We have revised and restructured our abstract to reflect the expected structure and the amendments in our article.
We have given our introduction a more logical order by themes. We removed material not directly related to our own data to make space for some of the other requested additions. The introduction now focuses on literature that also discusses the mental health of young people pre- and post-COVID-19, in particular the difference between LGBTQ+ young people and heterosexual cisgender young people, the difference between males and females and finally specific trends in the UK and Northern Ireland where our own study was situated.
We have added literature suggestions made by all reviewers if they fell under these themes if we had overlooked these in our own background research. Thank you for these suggestions.
We provided more specific details on our survey methodology, both the GHQ-12 instrument and why we used this to assess young people’s mental health, and the actual YLT survey, including methodological limitations. Here, we specifically provided more details on why the YLT survey is a suitable tool and why its sampling frame was the most suitable one for our study; we addressed the issue of sampling bias and dataset cleaning. We also provided more detail on how we used the qualitative data and how the open responses were coded, thematically organised and then chosen for inclusion.
The peer reviews were contradictory with regard to the need for an inclusion of a hypothesis and whether or not we ought to include policy recommendations. We discussed this and decided not to retro-fit a hypothesis, and therefore also largely maintain the structure of the conclusion. We did, as requested by some reviewers, strengthen the policy recommendations, in particular in relation to the need to include young people more in policy decisions and to improve mental health service provision for young people.
Specifically, we have, in line with your comments:
- added a comment on the potential response bias in open comments in surveys and generally tightened the methods section on the analysis of open-ended questions;
- We have proof-read the article again several times and double-checked the use of abbreviations;
- We double-checked that the referencing is correct;
- We noted your recommendations of related literature and have included your first and third recommendation.
Many thanks again for your valuable comments.
Round 2
Reviewer 2 Report
Dear authors,
Your article provides valuable insights into the impact of the COVID-19 pandemic on the mental health of young people in Northern Ireland. The findings contribute to the existing body of literature by highlighting the decline in mental health among 16-year-olds and identifying potential factors contributing to this decline. The inclusion of both quantitative and qualitative data strengthens the validity of your findings and enhances the comprehensiveness of the study.
The discussion section effectively connects the research findings with previous studies and global evidence, emphasizing the negative consequences of the pandemic on young people's mental health. Identifying key themes from the open-ended responses adds depth to the analysis and supports the quantitative data. Moreover, the limitations and strengths of the research are appropriately acknowledged, providing a balanced view of the study's scope and potential biases.
To further enhance the article, we suggest considering the following points:
Abstract
Refinement of Language: In a scientific abstract, you might want to avoid emotionally charged language such as 'felt acutely vulnerable'. Instead, use neutral language like 'reported significant concerns or challenges'.
Introduction:
Introduction of abbreviations: Ensure that all abbreviations are introduced before they are used in the text. For example, "Understanding Society (US)" was introduced well, but the abbreviation for "lesbian, gay, bisexual, transgender, and queer (LGBTQ+)" was not introduced before it was used.
Language and tone: Ensure academic and professional language throughout the text. Avoid contractions (e.g., use "has been" instead of "'s been") and be mindful of the tone.
Methods:
Ethical Approval: While ethical approval was obtained, participant confidentiality was not mentioned, especially considering the sensitive nature of some survey items.
Sample Bias: Although using the Child Benefit Register as a sampling frame is resourceful, it might also introduce some bias to the survey, as it excludes the children of higher earners. It's important to address how this sample bias might impact the results and interpretations.
results:
Clarity and Focus: Ensure that each paragraph maintains a focused discussion. For instance, in the social interactions and friends section, narrow your points to strictly issues related to friends and social interactions.
Author Response
Dear peer reviewer,
Many thanks again for looking over our article and for requesting further amendments, which we agree improve our article.
1. We have revised aspects of the article again to address these issues. We removed language that can be considered as emotive. The sentence you highlighted containing the phrase 'acutely vulnerable' now reads:
‘In the open responses, young people reported significant concerns about their mental health.’
2. We double-checked that all used acronyms are introduced properly in the correct places.
3. We have checked the manuscript and only contractions that were part of direct quotes from respondents remain. While we acknowledge that contractions should be avoided in academic writing, nonetheless, we feel the quotes should accurately reflect the language used by the young people.
4. You requested further information on confidentiality.
- We have added a sentence on data privacy/the privacy note that informed respondents about data protection and handling policy.The sentence in the Methods section reads:
In the information letter, 16-year olds were assured of confidentiality, and they received a link to our privacy notice, which sets out how participant data are used, stored and ultimately destroyed.
5. Our article had already addressed the potential bias re high earners. We had stated that high earners still receive Child Benefit payments, but that these are claimed back via income tax, which means that children of high earners are still contained in the child benefit register, so the sample bias is minimal. We've had lengthy discussions with HMRC at the time when the CB payment system was changed, and they assured us that the system remains de facto universal. However, we have added a further couple of sentences, stating that parents may choose not to apply for Child Benefit in which case their children would be excluded from the sample However, there could be various reasons why they do not apply, including, being a high earner, but equally mistrust in government r other reasons. We have added additional information to that effect. Again, we do know that the opt-outs do not in any meaningful way alter the sample, which is de-facto still universal, as we stated.
6. We have removed reference to playing sport and undertaking team activities in our para which focuses on spending time with friends, even though this is arguably where friendships are made and maintained. We checked all other paras, and they are strictly focused on one issue only.
Many thanks again for your comments.
Reviewer 3 Report
Dear Authors,
I appreciate the fact that you have improved the quality of your article, I have received sufficient clarifications although, in the Abstract, you have not yet defined the paragraphs related to Background, Material and Method, Results and Conclusions.
Success!
Author Response
Dear peer reviewer,
Many thanks for reminding us that we failed to insert the headers in the abstract of our article. This is now done in our final version, as requested.
Kind regards